# A Novel Improved YOLOv3-SC Model for Individual Pig Detection

**DOI:** 10.3390/s22228792

**Published:** 2022-11-15

**Authors:** Wangli Hao, Wenwang Han, Meng Han, Fuzhong Li

**Affiliations:** School of Software, Shanxi Agricultural University, Jinzhong 030801, China

**Keywords:** pig detection, YOLOv3, Convolutional Block Attention Module, Spatial Pyramid Pooling

## Abstract

Pork is the most widely consumed meat product in the world, and achieving accurate detection of individual pigs is of great significance for intelligent pig breeding and health monitoring. Improved pig detection has important implications for improving pork production and quality, as well as economics. However, most of the current approaches are based on manual labor, resulting in unfeasible performance. In order to improve the efficiency and effectiveness of individual pig detection, this paper describes the development of an attention module enhanced YOLOv3-SC model (YOLOv3-SPP-CBAM. SPP denotes the Spatial Pyramid Pooling module and CBAM indicates the Convolutional Block Attention Module). Specifically, leveraging the attention module, the network will extract much richer feature information, leading the improved performance. Furthermore, by integrating the SPP structured network, multi-scale feature fusion can be achieved, which makes the network more robust. On the constructed dataset of 4019 samples, the experimental results showed that the YOLOv3-SC network achieved 99.24% mAP in identifying individual pigs with a detection time of 16 ms. Compared with the other popular four models, including YOLOv1, YOLOv2, Faster-RCNN, and YOLOv3, the mAP of pig identification was improved by 2.31%, 1.44%, 1.28%, and 0.61%, respectively. The YOLOv3-SC proposed in this paper can achieve accurate individual detection of pigs. Consequently, this novel proposed model can be employed for the rapid detection of individual pigs on farms, and provides new ideas for individual pig detection.

## 1. Introduction

Pigs are the most common source of meat products worldwide. With the progress of human society, people pay more and more attention to the quality of pork.

Object detection technology has high value in improving animal welfare. Early in pig production, it can be utilized to monitor pig health to improve pork quality [1]. The dietary behavior of animals is closely related to their health status, and subtle dietary changes are important for animal health observations [2]. When pigs are sick, they usually show reduced feeding, reduced exercise, depression, and lethargy [3]. Leveraging scientific methods to monitor live pigs and, if necessary, human intervention, will help protect animal welfare and prompt pork quality and profitability. Initially, the vast majority of pigs were monitored manually, which led to significant increases in labor intensity. Meanwhile, during the monitoring process, human subjective judgment errors often occur, which was not conducive to the high-quality production of live pigs.

To handle the above problems, in the early days, researchers used RFID systems to monitor pigs’ diets. However, the sensitivity of RFID system monitoring was often affected by their surrounding environment as well as their own height, direction, and distance [4]. In pursuit of better accuracy, it is necessary to constantly adjust the position of the antenna [5], and RFID monitoring requires a large number of pig ear tags, which is time-consuming and expensive to maintain. Furthermore, there are also some problems to these approaches that depend on the utilization of the wearable equipment RFID such as being easy to damage, being invasive, and prone to infection [6,7]. Subsequently, the breeder deployed cameras to record the pigs’ behavior and manually analyzed the recorded video data, in order to obtain the health status of the pigs. These methods are all based on manual analysis, resulting in a significant increase in the workload of breeders.

Despite some desirable results, the above-mentioned methods suffer from compromised animal welfare and high physical labor intensity. This makes it urgent to leverage more efficient methods, for pig detection. The following are some effective attempts by researchers.

In [8], by leveraging elliptical displacement calculation methods, Kashila et al. has achieved 89.8% accuracy in pig movement detection. Concerning the individual pig classification, Kashila et al. [9] has received 88.7% accuracy in detecting individual pig identification via ellipse fitting technique. Based on traditional computer vision technology [10], Nasirahmadi et al. [11,12] has employed ellipse fitting and the Otus algorithm, to realize the individual pig detection and pig lying position detection. Furthermore, Nasirahmadi [13] has utilized support vector machine (SVM) algorithm to classify pig poses, with 94% classification accuracy achieved. Leveraging the linear discriminant analysis algorithm, Viazzi et al. [14] has achieved 89.0% accuracy in the recognition of aggressive behavior of pigs. Furthermore, a more promising method for individual identification and behavioral recognition of pigs is based on 2D or 3D cameras. For example, Matthews et al. [15] has utilized a depth camera to track the movement of pigs, enabling the effective detection of pigs’ standing, eating, and drinking behaviors. Depending on the depth sensor, Kim et al. [16] has realized the pig standing behavior recognition under the complex environment. Meanwhile, the effectiveness of the proposed method in terms of both the cost and the accuracy have been verified. Based on the images captured by the CCD camera, Nasirahmadi [11] et al. have utilized an ellipse fitting approach to locate each pig in the image, while cameras can easily record the pig behavior, factors such as farm environment and lighting conditions can make pig classification challenging.

Currently, deep-learning based approaches [17,18,19] have achieved promising detection performance, especially in the field of animal phenotype detection. For example, Wu et al. [20] has proposed an effective corbel detection method based on YOLOv3 and relative step size characteristic vector. Specifically, based on the relative step size characteristic vector, the YOLOv3 algorithm was utilized to detect the position of the corbel, and then the LSTM model was employed to identify the normal walking and the lame behavior of the cattle, and an accuracy of 98.57% obtained. Shen et al. [21] has first applied the YOLO model to detect cows, and then an improved AlexNet model has been employed to classify the corresponding detected individual cow. Finally, they obtained 96.65% accuracy of individual cow classification. Tassinari [22] proposed a deep learning-based system for individual cow classification and location analysis. Zhang [23] proposed a lightweight YOLO detection model, using MobileNetV3 to replace the backbone network in the YOLOv3 network, and obtained 96.8% of the cattle key position detection accuracy. Hu et al. [24] employed the YOLO algorithm to extract cow objects, and then a segmentation algorithm was utilized to extract the head, torso, and legs parts of the corresponding cow object. Subsequently, the deep feature fusion was performed on these extracted parts. Finally, the SVM classifier was employed to do the classification, and an accuracy of 98.36% was obtained. Jiang [25] proposed a filter-based YOLOv3 algorithm and achieved 99.18% accuracy in the detecting key parts of cows. Based on an RGB camera and convolutional neural network, Bezen [26] built a computer vision system for measuring cow feeding, and an accuracy of 93.65% was obtained. Achour [27] has built a CNN-based image analysis system for the classification of individual cows, their foraging behavior, and their food. Specifically, their model obtained an accuracy of 97% for individual cow classification and an accuracy of 92% for cow foraging behavior separately. Wu [28] proposed a CNN-LSTM (Fusion of Convolutional Neural Network and Long Short-Term Memory Network) model for cow action recognition. Specifically, the action categories of cows in their experiments included drinking, ruminating, walking, standing, and lying down, and the average classification accuracy of their model has reached 97.6%.

Above all, while the monitoring method using RFID ear tags is simple, it often causes harm to the pig and compromises animal welfare. Although the computer vision technology [29] can improve animal welfare and recognition accuracy, it is not suitable for industrial production requirements due to its slow detection speed. Furthermore, when the pigs are occluded, or the size of the target pigs in the image varies greatly, the detection performance of the model drops significantly.

Attention mechanism is an important means to improve feature robustness [30], among which Convolutional Block Attention Module (CBAM) [31] have shown promising success in a broad range of fields. Based on an intermediate feature map, CBAM captures attention maps in two independent dimensions, including channel and spatial dimensions. Then, the input feature map is multiplied with this attention map for adaptive feature refinement. Since CBAM is a general and lightweight component, it can be seamlessly incorporated into any CNN architecture for end-to-end training with negligible overhead.

Furthermore, the Spatial Pyramid Pooling (SPP) [32] module realizes the feature map-level fusion of local features and global features, enriching the expressiveness of the final feature map.

Consequently, based on YOLOv3, leveraging the advantages of CBAM and SPP, this paper proposes a novel improved pig detection model YOLOv3-SC. The YOLOv3-SC model enables efficient detection of pigs. In addition, the model can achieve effective pig detection in the case of occlusion, and can achieve effective multi-scale pig targets. The main contributions of this paper is summarized as follows:We first propose a novel pig detection method YOLOv3-SC based on the CBAM and the SPP modules. The channel attention and the spatial attention units in the CBAM module enable the YOLOv3-SC to focus on the regions of the image that are important for detection, thereby extracting richer, more robust, and more discriminative features. The SPP module endows YOLOv3-SC the capacity of extracting multi-scale features, which enables the model to detect objects of different sizes, thereby improving the model’s pig detection performance. Specifically, our model achieves the best performance for pig detection task, with 2.3% improvement of the existing model.Numerous ablation experiments have been designed and performed to verify the performance of our model. Specifically, these studies include the comparison of different models, evaluation of the effectiveness of the spp module, evaluation of the effectiveness of the CBAM module, and the evaluation of the superiority of the YOLOv3-SC.

## 2. Materials and Methods

### 2.1. Datasets

The individual pig detection dataset utilized in this paper was collected from one pig farm in Jinzhong City, Shanxi Province, China. The breeding method of this farm is captive breeding, surrounded by iron fences to form a closed area, and the ground of the farm is cement concrete. The data collection cameras were installed at a height of 3 m from the ground, at 45° diagonally and directly above the farm. Through this collection strategy, the whole view of the pig and its range can be well captured. The data collection period lasted for two months, from August to October 2020. It should be noted here that videos with poor picture quality are deleted due to factors such as light, and finally a total of about 2 Terabyte video data is obtained. Specifically, the video is sliced into image frames with rgb format at a sampling rate of 25 f/s and some images with no target objects, blurring and poor quality, are deleted. Further, the labelImg tool was employed to label the image frames in the PASCAL VOC format, and the labeled data was saved as an XML file. Finally, we obtained a dataset with a total of 4019 images with 13,996 annotations, some sample images are shown in Figure 1. Figure 1a indicates the camera position above the 45° angle of the farm. Figure 1b shows the camera position above the 45° diagonal of the farm. Figure 1c indicates that the camera position directly above the farm. In order to evaluate the performance of the proposed model, the dataset is divided as follows, 3255 samples are employed as the training data, the 362 samples are employed for validation data, and the remaining 402 samples are utilized as the test data. The samples in the test data are the unseen data. Further, to increase the diversity of data and allows the model to obtain richer features, this paper adopts the following data augmentation techniques, such as random scaling, random flipping, random cropping, and other operations.

### 2.2. Technical Route

The technical route of the individual pig detection model proposed in this paper is shown in Figure 2. To reduce the noise in the data and enable the model to obtain better detection ability, the samples are first preprocessed and data augmented. Specifically, the image preprocessing operation utilized for data noise reduction refers to deleting samples with poor quality in the data set and and resize the input image to a fixed size 416 × 416. In order to increase the diversity of data, we have adopted the following data enhancement methods, including the random_distort, random_expand, random_interp, and random_flip, shuffle_gtbox. Subsequently, the processed data is sent to the YOLOv3-SC for model training and evaluation. Finally, an effective individual pig detection model is obtained, which can realize fast and accurate individual pig detection.

#### 2.2.1. Feature Extraction

The backbone network DarkNet-53 with the addition of the CBAM of the proposed model, are utilized to extract features with richer spatiotemporal dependencies, which will facilitate the pig detection significantly.

#### 2.2.2. Feature Fusion

The Feature Fusion in Figure 2 refers to two kind of fusions; they are the multiple YOLO-head fusion and the SPP (Spatial Pyramid Pooling) fusion. The detailed feature fusion operations are described in Section 2.5.

### 2.3. YOLOv3

Based on YOLOv1 [33] and YOLOv2 [34], to leverage the anchor mechanism, BN operation, and multi-scale fusion strategies, we proposed YOLOv3 models. The basic principle of YOLOv3 is to divide the input image into S×S grids, where S=7 and each grid predicts 3 anchors. Each anchor has 5 parameters including (x,y,w,h,c), where *x* and *y* are the coordinate positions of the anchor, *w* and *h* represent the width and height of the anchor, and *c* is the confidence level of the predicted object. In addition to the parameters of the anchor, the YOLOv3 algorithm predicts the probability of each category and the confidence level can be achieved by the following Equation (Equation 1).
(1)Confidence=p_r(obj)×IoU
where p_r(Obj) is set as 0 or 1 and the IoU denotes the intersection ratio of the predicted and true frames. The confidence reflects whether the grid contains objects or not, and the accuracy of the prediction frame when the grid contains objects. Finally, the redundant anchor is eliminated by non-maximal suppression (NMS), and the position and size of the corresponding anchors are adjusted to produce the final result. YOLOv3 improves the detection performance via introducing the anchor mechanism based on YOLOv2 and a K-means clustering algorithm. The K-means clustering algorithm can be employed to obtain the suitable prior frame size, which is shown in the following Table 1 [35].

By introducing suitable prior frames, the network no longer needs to randomly generate anchor frames of different sizes to predict objects, thus making the network train faster and converge faster. YOLOv3 leverages a multi-scale strategy for object detection, which can detect more objects and identify smaller objects than those of YOLOv1 and YOLOv2, respectively. Concretely, the YOLOv3 network consists of four parts, including the input unit, backbone network unit, neck unit, and output unit. The backbone network in the YOLOv3 framework is the Darknet-53, and its basic unit is the residual structure [36], which can alleviate the gradient vanishing or explosion problems caused by the deepening of the network layers. The network structure of YOLOv3 is shown in Figure 3.

### 2.4. Attention Module

Pigs exhibit different actions at different times of the day, and there are often problems with pigs occluding each other. These situations can lead to a lack of distinct behavioral identification features in pig datasets. In order to handle the above-mentioned problems and improve the accuracy of the network by capturing more effective features, it is necessary for the network to learn action features adaptively. Consequently, this paper proposes an attention-enhanced YOLOv3 network, which aims to utilize the attention module to make the neural network pay more attention to the corresponding regions in the image, and these regions play a key role in action discrimination.

The Convolutional Block Attention Module (CBAM) [31] is a lightweight unit that consists of two separate parts, they are the Channel Attention Module (CAM, Channel Attention Module) and the Spatial Attention Module (SAM, Spatial Attention Module). CBAM is a combination of spatial attention and channel attention, which can be utilized to obtain rich semantic information in pig images. CBAM can capture the dependencies between channel feature space features, reduce the weight of unimportant information, and improve the detection performance of individual pigs. The structure diagram of the CBAM module is shown in Figure 4.

Specifically, CAM first performs a maximum pooling operation and a global average pooling operation on the input feature layer in turn, their output features are sent to a Multi-Layer Perception (MLP) layer. Then the output features of the MLP of two branches are summed together and send to a sigmoid function for fixing the weights between 0 and 1 distribution. The final result is obtained by multiplying the original input feature layer.

SAM first performs a maximum pooling operation and a global average pooling operation on the input feature layer, and then executes a tensor splicing on the corresponding output features. Finally, they are sent to the Sigmoid function to fix the weights between 0 and 1 distribution and then multiplied with the original input feature to achieve the final result.

The CBAM module can be represented by the following equation.
(2)$$F_{c}=Z_{c}\left(F\right)F$$
(3)$$F_{2}=Z_{c}\left(F_{c}\right)F_{c}$$
(4)$$Z_{c}\left(f\right)=\sigma \left\{F_{fc}\left[AvgPool\left(F\right)\right]+F_{fc}\left[MaxPool\left(F\right)\right]\right\}$$
(5)$$Z_{s}\left(f\right)=\sigma \left\{C_{c}\left[AvgPool\left(F1\right)\right]+F_{c}\left[MaxPool\left(F1\right)\right]\right\}$$
where $Z_{c}$ presents the channel attention module (CAM), $Z_{s}$ denotes the spatial attention module (SAM), *F* indicates the feature layer of the input network, and $F_{c}$/$F_{s}$ denotes the feature map after the Channel Attention Module (CAM)/Spatial Attention Module(SAM), respectively. × represents to perform the pointwise multiplication, and $F_{fc}$ is the fully connected operation. AvgPool denotes the global average pooling operation and the MaxPool indicates the global maximum pooling operation, respectively. $C_{c}$ presents the tensor splicing Concat operation. + represents the summation operation and the σ denotes the sigmoid activation function.

### 2.5. The Proposed Novel YOLOv3-SC Model

The proposed YOLOv3-SC model is built by leveraging the attention mechanism and Spatial Pyramid Pooling (SPP) module to the YOLOv3 backbone network Darknet-53. Specifically, the CBAM module endows YOLOv3 with powerful feature extraction capabilities. Furthermore, the SPP structure extracts features of different scales in the final stage of the backbone network and fuses them. This design can alleviate network overfitting, increase the robustness of the model, and allow the network to learn richer features. The architecture of the novel proposed YOLOv3-SC is presented in Figure 5.

Specifically, as in Figure 5 show, the YOLOv3-SC is enhanced by integrating the CBAM module in each Res_Block and adding SPP unit in the final stage of the backbone network.

The backbone network with the addition of the CBAM consists of Res1-CBAM, Res2-CBAM, Res4-CBAM, and Res8-CBAM. Res*i*_CBAM *i* = 1, 2, 4, 8 denotes a stack of CBL and *i* residual structures with CBAM modules, where CBL is composed of Conv, Batch Normalization (BN), and activation function (LeakyReLu) unit. The CBAM component allows the model to extract richer spatiotemporal features, which will facilitate the pig detection significantly.

Furthermore, the SPP structure is introduced for feature fusion. Specifically, SPP performs maximum pooling operations at different scales on the input feature maps, and finally all the output feature maps are tensor-spliced with the original feature maps. In this way, the network can perform feature fusion at different scales to prevent overfitting.

The input image is improved with the backbone network and SPP structure to obtain three different scales of feature layers with sizes of 13 × 13, 26 × 26, and 52 × 52. After that, further feature fusion operations are performed. The 13 × 13 feature map is upsampled to obtain the 26 × 26 feature map and the 26 × 26 feature map of the backbone network is concatenated. After that, we perform up-sampling again to obtain 52 × 52 feature map and 52 × 52 feature map in the backbone network for Concat operation to fuse the feature information of different scales. In the feature fusion stage, we first perform CBAM operation on the input of different feature layers, and then perform CBAM operation again after the tensor stitching Concat to obtain the improved feature fusion network.

The training process of the YOLOv3 is presented in Algorithm 1.
**Algorithm 1:** YOLOv3-SC Model Training
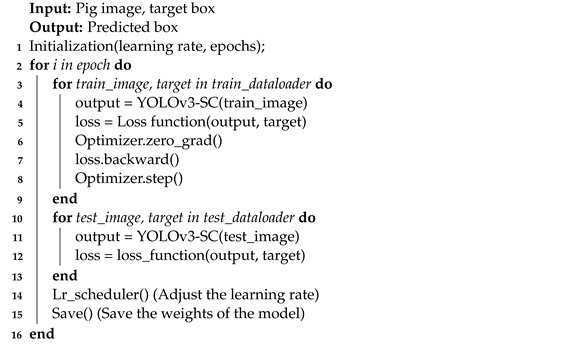


### 2.6. The Loss Function

In order to achieve the optimized pig detection model, a reasonable loss function needs to be designed for the network training. Specifically, the loss function consists of three terms: including category loss ($L_{cls}$), confidence loss ($L_{conf}$), and locality loss ($L_{loc}$). Among them, IoU is utilized to calculate the locality loss, and the cross-entropy loss is employed to calculate the confidence loss and class loss separately. The loss function of the proposed model is defined as follows.
(6)$$Loss=L_{loc}+L_{cls}+L_{conf}$$
where $L_{loc}$ indicates the error between the coordinates and the length and width of the real frame and the coordinates and length and width of the predicted frame, $L_{conf}$ illustrates the prediction region confidence error, and $L_{cls}$ denotes the object classification error, respectively.
(7)$$AP=\int_{0}^{1}P(R)dR$$
(8)$$mAP=\frac{\sum_{1}^{n}\left(AP\right)}{n}$$
(9)P=TP/(TP+FP)
(10)R=TP/(TP+FN)
where *TP* represents the number of positive samples predicted to be positive; *FP* indicates the number of negative samples predicted as positive samples; and *FN* illustrates the number of positive samples predicted as negative samples. *n* represents the number of the detected pig category, and its value is set as 1 here. *AP* is the average precision, denotes the area under the PR curve. mAP indicates the average accuracy over all categories.

### 2.7. Experiment Setup

In this paper, for fair comparison, all experiments are developed and run based on the PyTorch framework. The stochastic gradient descent optimization algorithm is employed for the model parameter update. Furthermore, the batchsize is 16, the initial learning rate is set as 0.001 and updated based on the cosine descent theory, the momentum is 0.937, and the total number of iterations is 400 epochs.

Additionally, the hardware configuration is as follows: operating system Ubuntu 20.04, CPU Intel(R) Xeon(R) CPU E5-2670 v3, GPU Nvidia Feforce GTX 3060 12G, and memory 16G DDR4.

## 3. Results

In this section, the experimental results and the discussions will be illustrated in detail. The experiments are organized in the following several parts, including the comparison of different models, the evaluation of the effectiveness of the SPP module, and the evaluation of the effectiveness of the SPP module. The purposes are to verify the effectiveness of the model, the effectiveness of the SPP unit in the YOLOv3-SC and the effectiveness of the CBAM unit in YOLOv3-SC correspondingly.

### 3.1. Comparison of Different Models

In order to validate the effectiveness of the proposed YOLOv3-SC, several models are utilized for comparison, including YOLOv1, YOLOv2, YOLOv3. Results are shown in Table 2.

Table 2 illustrates that the proposed model YOLOv3-SC achieves the best performance on all evaluation criteria. Specifically, YOLOv3-SC achieves 99.24% mAP, which is 2.31%/1.44%/1.18%/0.61% higher than that of YOLOv1/YOLOv2/Faster-RCNN/YOLOv3; the YOLOv3-SPP obtains 98.27% Precision, which is 4.54%/4.10%/3.53%/2.43% better than that of YOLOv1/YOLOv2/Faster-RCNN/YOLOv3; the YOLOv3-SPP achieves 94.31% Recall, which is 1.28%/1.44%/0.81%/0.22% superior than that of YOLOv1/YOLOv2/Faster-RCNN/YOLOv3; and the YOLOv3-SPP obtains 0.96 F1 score, which is 4.35%/3.22%/2.13%/1.05% higher than that of YOLOv1/YOLOv2/Faster-RCNN/YOLOv3. These all results validate the effectiveness of the proposed YOLOv3-SC.

The SPP structure and the CBAM attention component of the YOLOv3-SC allow the model to focus on the discriminant regions of the image in pig detection, fuse multi-scale feature maps, and extract more effective features even in the case of pigs sticking to each other. Consequently, YOLOv3-SC implements fast and efficient pig detection and achieves 99.24% mAP, which is significantly better than other models.

### 3.2. Evaluation of the Effectiveness of the SPP Module

In order to validate the effectiveness of the SPP module, we compare the YOLOv3 and YOLOv3-SPP. YOLOv3-SPP is built by encompassing the SPP module into the YOLOv3. Comparison results are shown in Table 3.

From Table 3, it can be seen that the YOLOv3-SPP model is superior to the YOLOv3 model on all evaluation criteria. Specifically, the YOLOv3-SPP achieves 99.19% mAP, which is 0.56% higher than that of YOLOv3; the YOLOv3-SPP obtains 97.19% Precision, which is 1.72% better than that of YOLOv3; the YOLOv3-SPP achieves 95.08% Recall, which is 1.02% superior than that of YOLOv3; and the YOLOv3-SPP obtains 0.96 F1 score, which is 1.05% higher than that of YOLOv3. These results validate the effectiveness of the SPP module.

The reason why the YOLOv3-SPP model is superior to YOLOv3 can be attributed to the following reasons. Specifically, the SPP module integrates both local and global features, thereby capturing multi-scale feature information, enhancing the expressiveness of features, and improving the robustness and the performance of the model.

### 3.3. Evaluation of the Effectiveness of the CBAM Attention Module

To evaluate the effectiveness of the CBAM attention module, some models are utilized for comparison, including YOLOv3 and YOLOv3-CBAM. YOLOv3-CBAM is established by leverage the CBAM module into the backbone of the YOLOv3. The comparison results are shown in Table 4.

Table 4 illustrates that the YOLOv3-CBAM model is superior to the YOLOv3 model on all evaluation criteria. Specifically, the YOLOv3-CBAM achieves 99.17% mAP, which is 0.54% higher than that of YOLOv3; the YOLOv3-CBAM obtains 97.46% Precision, which is 1.58% better than that of YOLOv3; the YOLOv3-CBAM achieves 94.24% Recall, which is 0.13% superior than that of YOLOv3; and the YOLOv3-CBAM obtains 0.96 F1 score, which is 1.05% higher than that of YOLOv3. These results evaluate the effectiveness of the CBAM module.

### 3.4. Evaluation of the Superiority of the YOLOv3-SC

To evaluate the superiority of the proposed model YOLOv3-SC, we compare it with the models in the above section, including the YOLOv3, YOLOv3-SPP, and YOLOv3-CBAM. YOLOv3-SC is built by integrating both the CBAM and the SPP modules into the structure of the YOLOv3. The corresponding comparison results are shown in Table 5 and Figure 6.

Table 5 and Figure 6 illustrate that the YOLOv3-SC model obtains the best results, which verifies the superiority of the integration of SPP and CBAM modules.

## 4. Discussion

In this paper, we propose an improved YOLOv3 model, YOLOv3-SC, to achieve the efficient detection of individual pigs. Specifically, in order to verify the superiority of the proposed model, this paper compares and discusses the performance of the four models YOLOv3, YOLOv3-SPP, YOLOv3-CBAM, and YOLOv3-SC. The experimental results demonstrate that both the YOLOv3-SPP and YOLOv3-CBAM models achieve better performance than those of the YOLOv3 model, which verifies the effectiveness of the SPP module and the CBAM unit. Moreover, the YOLOv3-SC model achieves the best performance, verifying the effectiveness of the proposed model. By leveraging the attention module CBAM, the proposed model can adaptively focus on the important features and reduce the weight information on the non-important features in pig detection. Furthermore, the SPP structure allows the model to combine the multi-scale information, which improves the model’s detection ability on small targets and adapts to the changing environment of individual pig detection in pig farms. The utilization of the SPP structure enhances the pig detection effect and performance of the model. Future work will explore more optimized data augmentation methods and more effective attention mechanisms that can be applied to more complex environments.

## 5. Conclusions

This paper develops a novel effective pig detection model YOLOv3-SC, which encompasses both the CBAM model and the SPP module into the backbone of YOLOv3 framework. The channel attention and the spatial attention units in the CBAM module enable the YOLOv3-SC to focus on the regions of the image that are important for detection, thereby extracting richer, more robust, and more discriminative features. The SPP module endows YOLOv3-SC the capacity of extracting multi-scale features, which enables the model to detect objects of different sizes, thereby improving the model’s pig detection performance. Ablation studies validate the superiority of both the CBAM and the SPP modules. Furthermore, experimental results show that the proposed YOLOv3-SC model obtains the promising pig detection performance. Specifically, the YOLOv3-SC achieves 99.24% mAP performance, which is significantly higher than those of the other popular models.

## Figures and Tables

**Figure 1 sensors-22-08792-f001:**
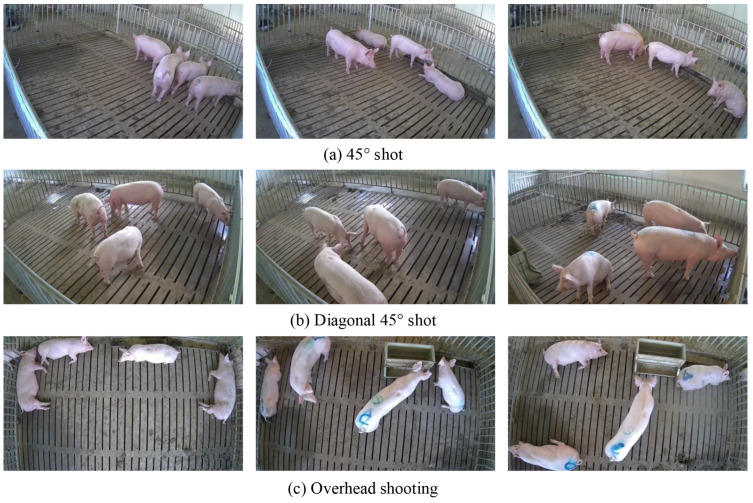
Some examples of the dataset.

**Figure 2 sensors-22-08792-f002:**
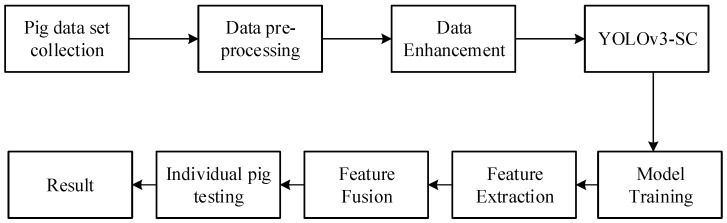
Technical route of the proposed pig detection model.

**Figure 3 sensors-22-08792-f003:**
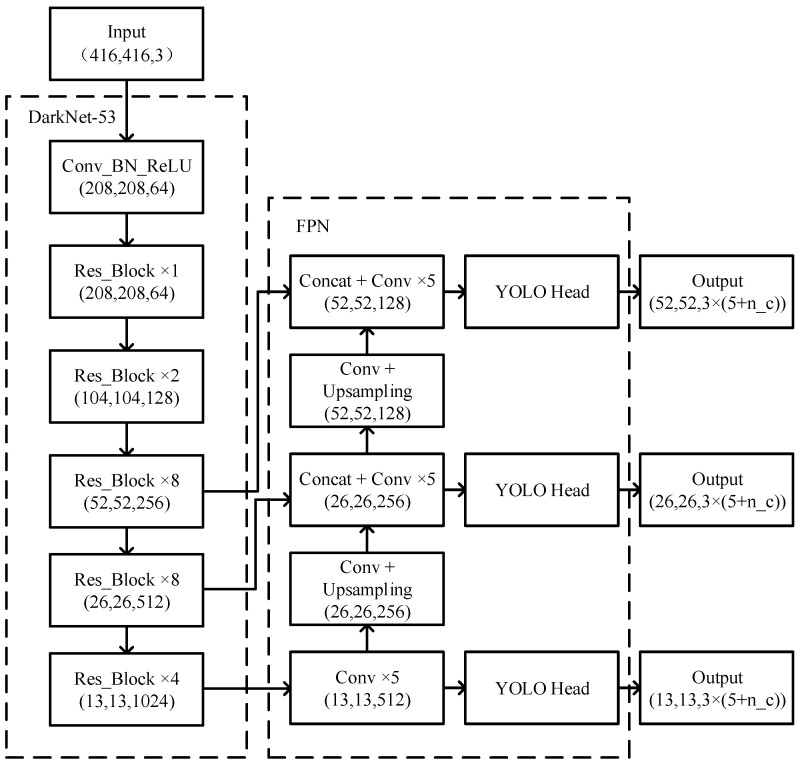
The network structure of YOLOv3.

**Figure 4 sensors-22-08792-f004:**
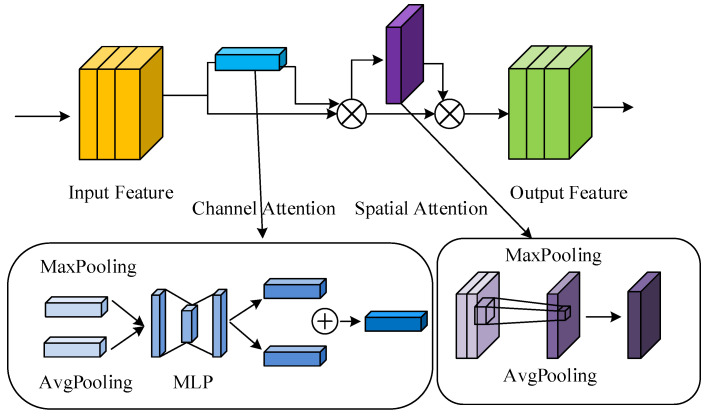
The structure of the CBAM module.

**Figure 5 sensors-22-08792-f005:**
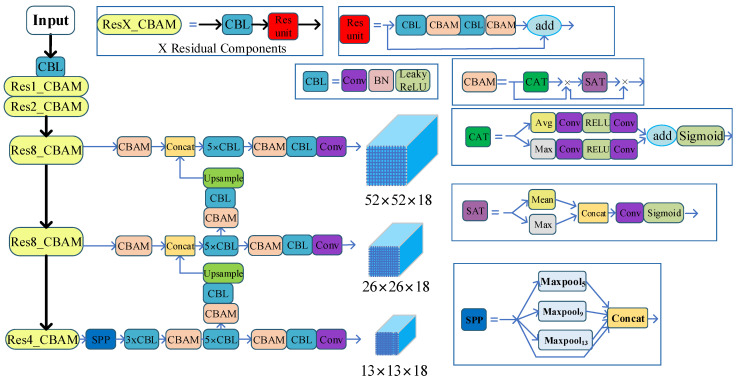
The pipeline of the proposed YOLOv3-SC. Here, we should note that the 5, 9, 13 in the Maxpool_5_, Maxpool_9_, Maxpool_13_ indicate the pooling kernel size of the maxpool operation.

**Figure 6 sensors-22-08792-f006:**
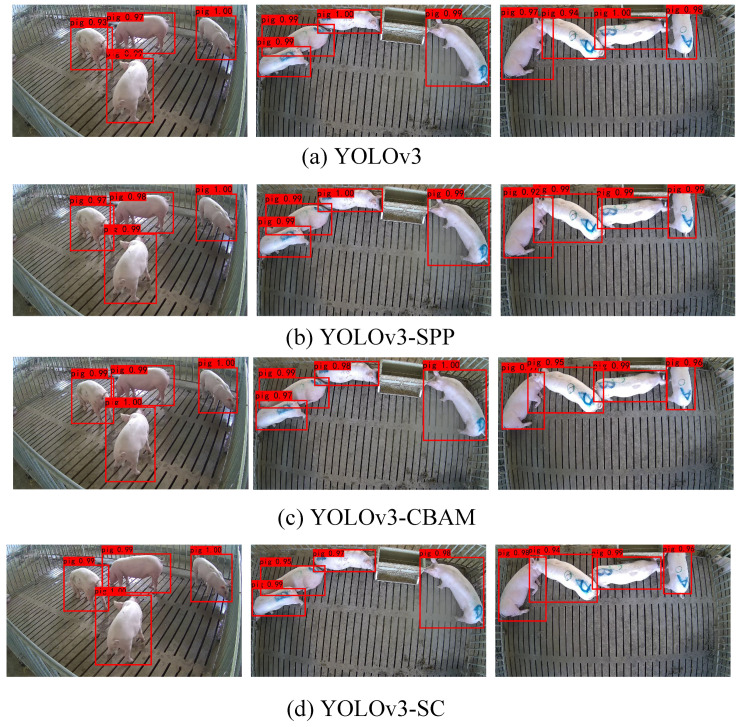
The detection results of different models.

**Table 1 sensors-22-08792-t001:** YOLOv3 Anchor Size.

Cell Size	Detection Box Size
13 × 13	(10,13) (16,30) (33,23)
26 × 26	(30,61) (62,45) (59,119)
52 × 52	(116,90) (156,198) (373,326)

**Table 2 sensors-22-08792-t002:** Comparison of YOLOv3 and YOLOv3-SC.

Model	Mean Average Accuracy (mAP/%)	Precision (P/%)	Recall (R/%)	F1 Score
YOLOv1	97.00	94.00	93.12	0.92
YOLOv2	97.83	94.40	93.55	0.93
Faster-RCNN	98.08	94.92	93.78	0.94
YOLOv3	98.64	95.94	94.12	0.95
YOLOv3-SC	99.24	98.27	94.31	0.97

**Table 3 sensors-22-08792-t003:** Comparison of YOLOv3 and YOLOv3-SPP.

Model	Mean Average Accuracy (mAP/%)	Precision (P/%)	Recall (R/%)	F1 Score
YOLOv3	98.64	95.94	94.12	0.95
YOLOv3-SPP	99.19	97.19	95.08	0.96

**Table 4 sensors-22-08792-t004:** Comparison of YOLOv3 and YOLOv3-CBAM.

Model	Mean Average Accuracy (mAP/%)	Precision (P/%)	Recall (R/%)	F1 Score
YOLOv3	98.64	95.94	94.12	0.95
YOLOv3-CBAM	99.17	97.46	94.24	0.96

**Table 5 sensors-22-08792-t005:** Comparison of YOLOv3-SPP and YOLOv3-CBAM with YOLOv3-SC.

Model	Mean Average Accuracy (mAP/%)	Precision (P/%)	Recall (R/%)	F1 Score
YOLOv3	98.64	95.94	94.12	0.95
YOLOv3-SPP	99.19	97.19	95.08	0.96
YOLOv3-CBAM	99.17	97.46	94.24	0.96
YOLOv3-SC	99.24	98.27	94.31	0.97

## Data Availability

The datasets generated during and/or analyzed during the current study are available from the corresponding author on reasonable request.

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
