# Peer review of "A Novel Improved YOLOv3-SC Model for Individual Pig Detection"

_sensors, 2022, doi:10.3390/s22228792_

Round 1

Reviewer 1 Report

Good job, keep improving.

Author Response

Q1:  Good job, keep improving

A1:  Thanks for your time and efforts devoted to the review of our paper, which are much appreciated.

Reviewer 2 Report

A novel improved YOLOv3-SC model for individual pig detection

1. Very interesting research entitled “A novel improved YOLOv3-SC model for individual pig detection”.

2. The structure of the article is correct. Only correct “3. Results and Discussion” by “3. Results” (line 275). See attached file.

3. Reference [27] is not cited in the article.

4. Figure 5 is not well appreciated, enlarge a little. Enhance the image.

5. Eliminate the paragraph of lines 125-127. It is not necessary to comment, what will be discussed later.

6. In the section “2.1. Datasets”, says that the pigs were captured with a video camera.  It remains to explain how the 4019 images used for processing were obtained (line 141).

7. In what image format were they taken (rgb, XYZ, L*a*b*, L*u*v*, HSV, HLS, YCrCb, YUV, I1I2I3, TSL, etc)?.   Was there any pre-processing of the images? Explain.

8. I suggest reviewing the following article:

Optimal color space selection method for plant/soil segmentation in agriculture. https://doi.org/10.1016/j.compag.2016.01.020

9. Of the 4019 images obtained: 3255 were used for training and 402 as test data. The sum of 3255 + 402 = 3657. What happened to the other images?

10. What is image preprocessing to reduce data noise (lines 151-152). Explain.

11. Develop an algorithm of the YOLOv3-SC model. I suggest that the algorithms in this article use the following format: See attached file.

12. Very good bibliography.

Note: The mentioned points in the review are recommendations from another perspective that gives the authors the opportunity to improve their paper. The questions are framed to propose a different standpoint and it is important that the authors address them and implement it in the entirety of the research study.

Reviewer 3 Report

Dear Editor and Authors,

Kindly refer to the attached file for feedback.

Regards

Round 2

Reviewer 2 Report

Thank you for making the indicated corrections.